# Compositionally Disordered Crystalline Compounds for Next Generation of Radiation Detectors

**DOI:** 10.3390/nano12234295

**Published:** 2022-12-03

**Authors:** Vasili Retivov, Valery Dubov, Ilia Komendo, Petr Karpyuk, Daria Kuznetsova, Petr Sokolov, Yauheni Talochka, Mikhail Korzhik

**Affiliations:** 1National Research Center “Kurchatov Institute”, Moscow 123098, Russia; 2National Research Center “Kurchatov Institute”—Institute of Reactives, IREA, Moscow 107076, Russia; 3Institute for Nuclear Problems, Belarus State University, 220030 Minsk, Belarus

**Keywords:** inorganic compound, crystalline material, scintillator, compositional disorder, light yield, scintillation kinetics

## Abstract

The review is devoted to the analysis of the compositional disordering potential of the crystal matrix of a scintillator to improve its scintillation parameters. Technological capabilities to complicate crystal matrices both in anionic and cationic sublattices of a variety of compounds are examined. The effects of the disorder at nano-level on the landscape at the bottom of the conduction band, which is adjacent to the band gap, have been discussed. The ways to control the composition of polycationic compounds when creating precursors, the role of disorder in the anionic sublattice in alkali halide compounds, a positive role of Gd based matrices on scintillation properties, and the control of the heterovalent state of the activator by creation of disorder in silicates have been considered as well. The benefits of introducing a 3D printing method, which is prospective for the engineering and production of scintillators at the nanoscale level, have been manifested.

## 1. Introduction

Scintillation materials play a notable role in ensuring further progress in the techniques and methods of detecting various types of ionizing radiation. Historically, their development has involved several phases driven by application. The nuclear projects in the 1940s of the 20th century led to the discovery of alkali halide scintillation single crystals. Later on, the drivers of the development were both scientific research in the field of nuclear and high energy physics and medical imaging [1]. Nowadays, security systems, environmental monitoring, and non-proliferation of fissile materials are the most significant industries that consume novelties in the field of scintillation materials. Unlike liquid, gaseous, and organic media, which are also used to detect ionizing radiation by the scintillation method, inorganic materials make it possible to combine high density, i.e., stopping power for ionizing radiation, with a wide variation in spectral and luminescent parameters due to the choice of material composition and activating ions. The widespread application of modern analytical methods for studying materials at the submicron level [2,3,4] and measuring electronic excitation transfer processes in the subpicosecond range [5,6] made possible the development of materials at a new level. It is worth noting the progress in the development of the theory of scintillations [7,8], and the use of big data analysis of spectroscopic data [9,10,11,12,13,14,15,16] to identify patterns in scintillation properties when varying the composition of relatively simple compounds. The accumulated experience and knowledge in this area, as well as the available tools, have made it possible to proceed with the targeted engineering of more complex compounds at the submicron level.

Since the publication of the second edition on *Inorganic Scintillators for Detectors Systems* by P. Lecoq et al. [17], quite popular among specialists, five years have passed. During this time, new directions in the creation of scintillation materials and detector elements have been formed. Particularly noteworthy are structured materials and detector elements [18,19,20,21,22], as well as the incorporation or creation of nanoparticles of alkali halide materials with a high scintillation yield in various media, in order to obtain new properties and improve the manufacturability of these materials [23]. These developments put forward the work with nanoscale objects for luminescence generation high on the list of priorities. Nonetheless, the investigation of inorganic scintillation materials in a variety of forms, from nanoparticles to bulk single and polycrystalline ceramic blocks, as well as films [24], remains at the heart of this field’s innovations. Therefore, the progress in the luminescence properties of the inorganic materials remains an important task for the creation of detector elements that meet the ever-increasing requirements for operational characteristics: energy and time resolution and tolerance to various types of ionizing radiation.

Analyzing the properties of the most commonly used scintillators, it can be stated that the greatest progress in improving their properties was achieved through the use of codoping: LaBr_3_: Ce [25], NaI:Tl [26,27], CsI:Tl [28], PbWO_4_ [29]. Another way to improve the consumer performance is the use of the sampling of detector elements with the best parameters among those produced, which significantly increases the cost of final products.

The main limiting factor for alkali halide compounds is the isolation of all technological stages of production from an oxygen-containing atmosphere. The incorporation of oxygen ions into the crystal lattice, in particular, based on Br- and I-anions, leads to the formation of point defects competing for trapping nonequilibrium carriers while the material interacting with ionizing radiation. These issues are the limiting factors for the application of the new generation halides LaCl_3_:Ce and LaBr_3_:Ce [30,31] in widely used pixelated detectors in medical scanners.

A limiting factor for self-activated scintillators of the bi-cation tungstate family and materials of the olivine structural type is the temperature quenching of luminescence and, as a result, the low scintillation yield, which cannot be increased by technological methods. The only example is a twofold increase in the scintillation yield in a PWO-II crystal. However, it was achieved by reducing the concentration of the doping of La^3+^ and Y^3+^ ions. Such combining doping is used to stabilize the radiation hardness of the material.

In Ce-activated oxide compounds, investigation has focused on the search for scintillators with a combination of fast kinetics, high density, and high scintillation yield [32]. Besides orthosilicates [33], binary compounds with garnet Lu_3_Al_5_O_12_: Ce(Pr) (LuAG) [34,35,36,37] and oxyorthoaluminates YAlO_3_ (YAP) and Lu(Y)AlO_3_ (LuAP) [38,39,40] structures activated by Ce^3+^ and Pr^3+^ ions were discovered.

Among the listed oxide scintillators, materials with a garnet structure have the greatest potential for improving scintillation properties. We note three important features of these compounds:

First, compounds of the garnet structural type have a cubic space group O^h^_10_-Ia3d [41]. The isotropy of the matrix properties allows the use of many technological methods for shaping, annealing, and growth in order to obtain a high-quality crystalline mass. They have a good optical transparency, which can be varied over a wide range by the composition of matrix-forming and doping elements [42,43,44,45].

Second, the isomorphism of garnets is a unique opportunity to obtain new compounds of this class. Their general formula can be represented as C_3_B_2_A_3_O_12_, where the C, B, and A positions correspond to cations in different crystal positions: dodecahedral, octahedral, and tetrahedral, respectively [46]. Ions of Ce^3+^ and other rare earth (RE) ions are predominantly localized in dodecahedral positions.

Thirdly, the isovalent localization of RE activators in the above positions in the structure is accompanied by a change in the corresponding metal–oxygen bond length. This affects their luminescent properties. This mechanism is an additional tool for adjusting the spectral-luminescent properties of the garnet type compounds. It should be noted that the garnet lattice can be formed both by pairs of di-and tetravalent ions and by pairs of trivalent ions. A heterovalent substitution in garnets is also possible, but it is limited by the necessity for charge compensation [47,48].

In the family of the artificial garnets, compounds based on trivalent ions are the most widely used. Compounds with a garnet structure can be expressed as C^3+^_3_[B^3+^]_2_(A^3+^)_3_O_12_, where C = Y, Gd, Tb, Er, Tm, Ho, Yb, Lu; B, C = Al, Ga, Sc, etc.

One of the most promising compositions for combining high scintillation parameters and high stopping power was lutetium garnet [49]. However, from the experience of the technological development of the binary Lu garnet crystal, a few problems have been disclosed, which have become a significant limiting factor. Note the low, no more than 1 mm/h, rate of pulling out the commercial crystalline mass on the seed, which leads to a gradient in the properties of single crystals along the crystal axis. Moreover, due to a difference in segregation coefficients of various ions from the melt at a low pulling rate, an axial gradient of defects arises, which significantly limits the volume of the crystalline mass suitable for manufacturing the high-quality detector elements. Finally, in the single crystal with a garnet structure, when a small-radius Lu^3+^ ion is included in the composition, antisite defects occur. The lutetium ion is localized in dodecahedrons and, in a small quantity, in octahedrons as well [50,51].

Technologies to produce ceramic materials are devoid of these disadvantages. Moreover, the capability of strict control at the production of precursors (powders for further densification) provides better scintillation characteristics of YAG:Ce [52], LuAG:Ce [53], LuAG:Pr [54] ceramic scintillators than their single-crystal counterparts due to the provision of homogeneous distribution and increased dopant concentration, as well as the use of technological methods to reduce the concentration of antisite defects.

Notable progress in improving the scintillation yield of LuAG:Ce ceramics has been made in just a decade. LuAG:Ce,Mg ceramics showed a high yield of 21,900 ph/MeV with a signal integration time in a time gate of 1 µs [55], and a yield of 25,000 ph/MeV has been obtained so far. An even more significant effect of improving the scintillation parameters was observed when the LuAG:Ce ceramic was codoped with Li^+^ ions. LuAG:Ce,Li ceramics demonstrated a scintillation yield of 29,200 ph/MeV [56].

Additional improvements in the properties of LuAG:Ce ceramics are achieved by adjusting the composition of the precursor, which provides a variation in the composition near the stoichiometric ratio. The Lu^3+^ deficient nonstoichiometric LuAG:Ce ceramics showed poor optical transmission due to the presence of a secondary phase but improved scintillation performance [57,58].

Thus, it can be stated that intensive research on improving the scintillation properties of binary crystalline garnets, although it resulted in some progress, still did not allow reaching the set of scintillation parameters better than the lutetium oxyorthosilicate Lu_2_SiO_5_:Ce family owes. Similarly, one can also be declared that there is a stagnation in the breakthrough increase in scintillation yield in simple (two-component) alkali halide materials with various activators.

This review is devoted to the analysis of the compositional disordering potential of the crystal matrix of a scintillator to improve its scintillation parameters. The effects of the disorder on the landscape at the bottom of the conduction band, which is adjacent to the band gap, have been discussed. The ways to control the composition of polycationic compounds when creating precursors, the role of disorder in the anionic sublattice in alkali halide compounds, and a new kind of disordered material on a base of silicates have been considered as well. The benefits of introducing a 3D printing method, which is prospective for the engineering and production of scintillators at the nanoscale level have been argued.

## 2. Next Step: Compositional Disordering of the Materials

Let us turn to the analysis of a simplified expression describing the scintillation yield (LY) [7]. It usually includes three factors: LY = ηSQ, where η describes the efficiency of ionizing radiation energy conversion; S is the efficiency of nonequilibrium carrier energy transfer to an ensemble of luminescent centers; and Q is the quantum yield of intracenter luminescence of these centers. The first is exclusively due to the properties of the matrix, in particular, the size of the band gap which determines the energy of thermalized carriers and, accordingly, excitons. The latter is defined by the properties of the luminescent center. For intracenter spontaneous transitions in a distinct luminescent center, Q may exceed unity. For example, relaxation from the radiative state of Ce^3+^ ions can be accompanied by the emission of an optical photon upon transition to the ^2^F_7/2_ level and then by the emission of a photon with an energy of ~2000 cm^−1^ in the far IR region when thermodynamic equilibrium is established between the ^2^F_7/2_ and ^2^F_5/2_ states. However, photodetectors used in ionizing radiation measurement are typically not sensitive in the far infrared spectrum. The value of Q can exceed unity upon excitation in polyatomic systems when the excitation energy of one of the participants in the cooperative process is divided between neighboring centers emitting the same spectrum [59,60,61,62,63].

The efficiency of energy transfer S is determined by the conditions for the transfer of both bound nonequilibrium carriers (excitons) and the migration rates of electrons and holes in the conduction and valence bands, respectively. The effective mass of holes in the valence band in dielectrics is, as a rule, an order of magnitude or even greater than the mass of electrons in the conduction band. Therefore, the migration of electrons along the band, as well as the mobility of excitons, determine the efficiency of excitation of luminescent centers. In binary systems, which typically have the regular shape of the crystal potential forming the bottom of the conduction band, there is a significant expansion of nonequilibrium electrons from holes. This prevents the creation of excitons and, consequently, the low efficiency of the excitation transfer through the excitons [8]. This lack causes the low contribution of this mechanism to scintillation, for example, in binary oxide compounds. The reduction in the expansion of nonequilibrium carriers can be facilitated by modulation of the crystal potential at the atomic level, which may be created by introducing a compositional disorder into the matrix.

Compositional disorder in a crystalline system does not mean amorphization of a compound. A polycationic ensemble of the cationic sublattice with identical crystallographic sites creates a disordered distribution of cations in the matrix. However, the conservation of the space symmetry group occurs due to the ordered distribution of anions. Thus, the long-range order in the ion stabilization positions, i.e., crystallinity, is preserved. This approach is typical for oxygen systems. For alkali-halide systems, where anions may also be mixed, the long-range order is preserved due to the ordering of the cationic subsystem. Highly disordered crystalline systems consisting of many cations occupying the same positions in the crystal lattice are often referred to as high-entropy materials [64,65]. They also find application in photocatalysis [66] and the creation of more complex systems [67] than those that have been discussed in this review. In [68], promising high-entropy ceramics have been considered specifically for photonics applications as well.

Apparently, research on oxyhalides with RE activators [69] and Gd_2_O_2_S:Tb [70] may be considered as a pioneer in the creation of compositionally disordered systems in the field of scintillators. Later on, an improvement of the technological capabilities for single phase stabilization and better scintillation properties by growing a mixed perovskite-type Lu_x_Y_1-x_AlO_3_:Ce instead of LuAlO_3_:Ce and oxyorthosilicate type (Lu_x_Y_1-x_)_2_SiO_5_:Ce (LYSO) instead of Lu_2_SiO_5_:Ce (LSO) demonstrated [71,72,73].

The most technologically advanced combinations for introducing compositional disorder into a garnet-type crystal system are either the partial isomorphic substitution of aluminum ions by gallium ions or the mixing of isovalent cations localized in the dodecahedral position. There are other types of disorder in crystalline systems [74], but for the development of scintillators, compositional disorder is of particular interest.

### 2.1. Conduction Band Bottom Landscape Modulation

The difference in the formation of the spatial distribution at the bottom of the conduction band might be revealed by considering fluctuations in the effective potential in the supercell of a mixed crystal with a random cation distribution. Fluctuations of the effective potential in the single-electron approximation, causing the scattering of electron states of the first branch in the conduction band of the crystal, can be built using the pseudopotential method according to the model proposed in [75]. The pseudopotential landscape can be expressed as:(1)$$u\left(\vec{r}\right)=\sum_{\vec{R}}p_{\vec{R}}\Delta E_{cb}\frac{4}{\pi }\frac{V_{0}}{{\left(\kappa a\right)}^{3}}s\left(\frac{\left|\vec{r}\right|}{\kappa a}\right),$$
(2)$$s\left(x\right)=\{\begin{array}{c} \begin{array}{c} 0,\;\left|x\right|>1\;\\ 2{\left(1+x\right)}^{2},-1\leq x<-0.5 \end{array}\\ \begin{array}{c} 1-2x^{2},-0.5\leq x<0.5\\ 2{\left(1-x\right)}^{2},\;0.5\leq x\leq 1 \end{array} \end{array}$$
where *x* is independent variable of the function *s*(*x*), $p_{\vec{R}}$ is the matrix of nodes occupied by Al (Y) ions in a supercell, $\Delta E_{cb}$ is the difference between bottoms of the conduction bands of two components of a solid solution, $V_{0}$ is the volume of the unit cell, a is the lattice constant, κ is the scale factor. The scale parameter means the distance (in units of a lattice constant) at which the pseudopotential of a certain substituting ion equals zero.

Figure 1 presents a 2D cross-section of the spatial distribution of the pseudopotential $u\left(\vec{r}\right)$ in binary crystals: Y_3_Al_5_O_12_ and Y_3_Ga_5_O_12_, Lu_3_Al_5_O_12_ and Lu_3_Ga_5_O_12_, Lu_3_Al_5_O_12_, Y_3_Al_5_O_12_, Lu_3_Gd_5_O_12_ and Y_3_Ga_5_O_12_, as well as ternary crystals having the same content of Al^3+^ and Ga^3+^ or Lu^3+^ and Y^3+^ ions, respectively. The spatial distribution of the substituting ions was chosen to be homogeneous, without clustering. The 2D cross sections are built in 15 × 15 × 15 supercells for garnets and in 20 × 20 × 20 supercells for oxyorthosilicates at $z=\frac{15}{2}{\left(\vec{a}+\vec{b}+\vec{c}\right)}_{z}$, i.e., at the half height of the supercells. The parameters of the unit cells of binary components, the positions of ions in the unit cells and the band gaps are obtained from [76,77,78,79,80,81].

The amplitude of the fluctuations of the pseudopotential reaches the largest values in a ternary system composing Al^3+^ and Ga^3+^ ions. It is caused by the largest difference in the band gaps of the binary compounds Lu_3_Ga_5_O_12_ and Y_3_Ga_5_O_12_ or Lu_3_Al_5_O_12_ and Y_3_Al_5_O_12_. At the same time, the substitution of yttrium by lutetium ions leads to a less modulation of the effective potential. A smaller effect is observed when yttrium is substituted by lutetium in the Lu_2_SiO_5_-Y_2_SiO_5_ system as well. Figure 2 presents a minor modulation of the effective potential in (L_0.5_-Y_0.5_)_2_SiO_5_ solid solution.

The effects of the modulation of the effective potential in a disordered crystal on the mobility of nonequilibrium carriers can be classified into two groups: elastic scattering of free carriers with the kinetic energy above the localization threshold on potential fluctuations, and localization of thermalized carriers in potential wells with the possibility of subsequent activation due to phonon absorption or tunneling into a neighboring potential well [82]. The latter effect has a lower probability in comparison with the thermal activation of carriers at room temperature. In both cases, the modulation of the effective potential leads to a decrease in the diffusion length of free carriers, increasing the concentration of geminate electron-hole pairs, and, consequently, increasing the transfer efficiency to the luminescent centers.

Eventually, the modulation of the effective potential changes the energy structure of single-electron states in the crystal. The interruption of the periodicity of the crystal field leads to a spatial compression of states. In such systems, the wave vector is already a “bad” quantum number. Unfortunately, such quantum states cannot be described by any set of quantum numbers. However, such mixed systems can be considered in the context of a virtual (average and regular) crystal with a finite lifetime of single-electron states due to scattering on fluctuations of the crystal field, as described in [75]. Such an approach uses the coherent potential (CPA) method [82] to calculate the band structure in a mixed solid solution. The branches in the conduction band of such mixed crystals are broadening, which width corresponds to the uncertainty of the energy of the states.

The branch broadening changes the shape of the density of states, which acquires a tail in the energy region below the edge of the fundamental absorption in the virtual crystal $E_{g}^{vc}\approx xE_{g}^{A}+\left(1-x\right)E_{g}^{B}$, where $E_{g}^{A}$ and $E_{g}^{B}$ are the band gaps of binary components. The broadening of the lower branch in the conduction band at the neighborhood of the Γ point can be considered as the blurring of the bottom of the conduction band. The blurring of the bottom of the conduction band is very often interpreted as a change in the width of the band gap. This approach is based on linear absorption spectroscopy results when the dynamic range of the setup does not exceed three decades, and blurring causes the observed shift of the cutoff absorption.

The significant predominance of the concentration of one of the cationic components, which generates the disorder, might lead to clustering in the compound, i.e., the appearance of regions of individual compounds with the same structural type [83], and the scintillation parameters of such materials are quite close to the parameters of initial components in a series of solid solutions.

### 2.2. Changing the Parameters of the Local Crystal Field

A change in the relief of zones also affects the local crystal field, causing a strength variation in the Stark splitting of activator levels [84,85]. Disordering in the Al-Ga sublattice to a greater extent, and in the Y(Lu)-RE sublattice to a lesser extent, leads to fluctuations of the crystal field in the localization position of the Ce^3+^ activator ions. In the crystal lattice of garnet, dodecahedrons containing Ce^3+^ ions are surrounded by tetrahedra and octahedra occupied by Ga^3+^ and Al^3+^ ions. The strongest effect on the 5d states of the Ce^3+^ ion can be expected from two tetrahedra sharing d_48_ edges with the dodecahedron containing Ce.

In a garnet-type lattice, Ga^3+^ ions tend to occupy tetrahedral positions [86,87]. The population of the tetrahedra with Ga^3+^ or Al^3+^ ions lead to substantially nonequivalent positions for Ce^3+^ ions. It has been experimentally established that the length of common edges between the dodecahedron and the tetrahedron d_48_ plays a decisive role in the splitting Δ_1,2_ between the states 5d_1_ and 5d_2_ [88]. The length d_48_ is longer in the tetrahedra occupied by Ga^3+^ than in the tetrahedra occupied by Al^3+^ ions. The greater the d_48_, the smaller the Δ_1,2_. This contributes to a decrease in the energy gap between the 5d_1_(Ce^3+^) radiative state and the modulated bottom of the conduction band, while the opposite trend is observed at a predominant concentration of Al^3+^ ions in the compound composition. LuGAG:Ce single crystals exhibit a lower slow scintillation component and a higher LY value compared to LuAG:Ce [89,90]. However, with an increase in the Ga content due to a decrease in the strength of the crystal field in the dodecahedral position, the energy level 5d_1_ of the Ce^3+^ activator shifts towards higher energies, while the average crystal potential shifts towards lower energies, approaching each other. This reduces the energy of thermal ionization of Ce^3+^ ions into the conduction band, makes the process of thermal ionization more probable at room temperature, and, as a result, leads to a slowing down of the scintillation kinetics. A similar change in parameters was also found in LuGAG doped with Pr^3+^ ions [91].

The effect of shallow electron traps caused by compositional disordering can be diminished by codoping with a divalent ion. The effect of codoping in Y(Lu)-aluminum-gallium garnets turned out to be similar to the effects in binary compounds with a garnet structure. Furthermore, codoping with Ca^2+^ and Mg^2+^ ions affects the concentration of vacancies and retrapping processes in disordered systems, which influences the scintillation characteristics [92,93]. At a high concentration of codoping, they can contribute to the localization of tetravalent cerium ions in the lattice. Ce^4+^ can act as a fast nonradiative relaxation channel, and thus suppress the scintillations caused by Ce^3+^. However, at a low concentration of Mg^2+^ ion codoping in disordered compounds of both oxyorthosilicates and garnets, a broad structureless absorption band with a maximum near 260 nm appears in the optical absorption spectrum [94]. The intensity of this band does not show a change with time. Therefore, it is not associated with a metastable color center. It is most likely due to a charge–transfer transition from the top of the valence band to the unfilled level of the Mg^2+^-based defect. No luminescence bands associated with excitation into this band were found, which indicates a very fast nonradiative relaxation. This means that, during the formation of scintillation, carriers trapped in shallow traps due to matrix disorder having a thermal activation energy of the order of fluctuations of the crystal potential (>0.5 eV) will be predominantly recaptured by the magnesium impurity-based defects. For this reason, their delocalization into the conduction band will be minimized, resulting in faster scintillation kinetics by Ce^3+^ ions. An inhomogeneous broadening of the luminescence of Cr^3+^ ions and a change in the local symmetry of the activator site were noted in LuYAG [95]. Compositional disordering leads to inhomogeneous broadening of the luminescence band of Ce^3+^ ions and, consequently, to a gradient in the luminescence kinetics along the band contour [96]. The transition from the ternary compound Gd_3_Ga_3_Al_2_O_12_:Ce to the quaternary (Gd,Lu)_3_Ga_3_Al_2_O_12_ leads to even greater inhomogeneity of the luminescence kinetics, as can be seen from Figure 3.

Thus, the creation of compositional disorder in the Al-Ga and Y-RE systems in the garnet structure leads to the possibility of controlling the spectral-luminescent properties of the scintillator and the position of the emitting level of Ce^3+^ ions relative to the modulated bottom of the conduction band, which makes it possible to purposefully adjust the luminous properties of the material.

### 2.3. Modulation of Local Charge Distribution

Compositional disorder, in addition to a modulation of the crystal potential, can provide a local fluctuation of the charge. Recently, new results on phosphors [97,98,99] were announced. Attempts have been made to create materials with a garnet structure that combine di-, tri-, and tetravalent cations in a matrix. Such an approach promises certain benefits in further reducing the scattering length of nonequilibrium carriers formed by ionizing radiation. However, the development in this direction is hindered by technological factors associated with the difficulties in controlling the charge state of activators in such systems.

### 2.4. Variation in Stopping Power and Sensitivity to Various Types of Ionizing Radiation

The compositional disorder in the sublattice of a heavy cation makes it possible to solve the problem of adjusting the stopping power to various types of ionizing radiation. The compound stopping power for hard electromagnetic radiation depends on the density and the effective charge Z_eff_ of the compound as ~Z^4^_eff_. It is enough to have a light Y cation in the dodecahedral position in the matrix to detect soft X-rays. To detect higher-energy gamma quanta, especially annihilation quanta with an energy of 511 keV, most typical in PET scanners, this position should be filled with heavier cations like lutetium and gadolinium. By controlling the compositional disorder in the dodecahedral position, the density of the compound can vary from 4.55 (YAG) to 7 (LuAGG) g/cm^3^. To detect hard gamma quanta, when the formation of e^−^–e^+^ pairs become dominant, the radiation length X_0_ of the garnet material can be brought to values of less than 2 cm by filling dodecahedral positions with heavy cations. This makes it possible to use garnet-type materials to create compact calorimetric detectors in high-energy physics. The variability of the stopping power of the garnet compound by controlling the composition in dodecahedral positions can bring the partial ionization loss for high-energy charged minimally ionizing particles (MIP) to a level of 0.75 MeV/mm, which is five times higher than for scintillators based on organic plastics. This makes it possible to create detectors with increased time resolution for collider experiments.

The use of Gd to fill the dodecahedral positions allows one to solve another important detection problem—the detection of neutrons by the scintillation method. Natural gadolinium is in the form of a mixture of six isotopes, two of which, ^155^Gd and ^157^Gd, have the highest cross section of interaction with thermal neutrons of all known isotopes, 61 and 255 kbarn, respectively.

Figure 4 shows the part of absorbed neutrons (in percent) in gadolinium layer thicknesses of 0.2–10 mm of natural isotopic composition for incident neutron energies in the range from thermal 0.0253 eV to fast 15 MeV. Since crystalline compounds with a garnet structure have a density of gadolinium nuclei not much less than that of a metal, it is obvious that relatively thin layers of a crystalline compound, less than 5 cm, can provide effective absorption of neutrons in a wide energy range. Gadolinium nuclei possess a wide range of resonances for neutron absorption in the energy range from 1 eV to 10 keV, which makes them effective for detecting epithermal neutrons.

The products of the neutron capture reaction are γ-quanta and conversion electrons with a total energy of ~8 MeV. At the radiative capture of neutrons by gadolinium nuclei, compound nuclei are formed in an excited state, which excitation relaxation is accompanied by the emission of a cascade of γ-quanta, the average number of emitted gamma-quanta is four at an energy of detected neutrons of 0.0253 eV [100,101]. For detection, the most interesting are the intense and highest-yielding soft lines of gamma radiation with energies in the range of 80–90, 180–200 and 511 keV. Control of sensitivity to various types of ionizing radiation can also be achieved in alkali halide compounds, where Li and Na ions are isovalently mixed in the (Na,Li)I:Tl compound [102]. It could also be achieved by eutectic compound [103]. Eutectic Ce:^6^LiBr/LaBr_3_ with a high Li concentration was produced via the vertical Bridgman method and exhibited a lamellar eutectic structure and optical transparency. The light yield under neutrons excitation was estimated to be 74,000 photons/neutron. The scintillation decay time was defined to be 18.7 ns.

## 3. Producing and Utilizing of the Precursors

### 3.1. Development of Nanosized Powders

With the increasing complexity of the elemental composition of the material, the quality of precursors for crystal growth and for ceramics production comes to the fore. A precursor is a mixture of compounds suitable for further thermal treatment or compacting, with a minimal deviation from a given composition, usually stoichiometric, during further treatment (melting, sintering). Obviously, the precursor that provides the highest rate of transformation into a commercial crystalline mass is of the greatest interest. The precursor can be obtained in a relatively simple way by mechanical mixing and milling of powders for further solid-phase synthesis [104,105]. Furthermore, more complex methods such as co-precipitation [106,107], sol-gel [108,109], and spray-pyrolysis [110,111] are used.

For gallium-containing mixtures, a characteristic process is the evaporation of gallium oxide [112] during heat treatment. During single crystal growth, Ga_2_O_3_ evaporation is minimized by using a slightly oxidizing atmosphere (e.g., Ar/O_2_) [113]. In the case of ceramics, a known way to solve this problem is sintering in an oxygen atmosphere [114]. The listed methods for the synthesis of precursor powders of a given stoichiometry were used to obtain single crystals and ceramics. Authors [115] demonstrated that nanostructuring plays a positive role in the production of complex compounds. Precursors for transparent ceramics of binary and ternary garnets were obtained by solid phase synthesis as well as by co-precipitation and spray pyrolysis [116,117,118,119,120,121]. Among these methods, co-precipitation has an advantage of the fact that the necessary crystalline phase has already formed during the preparation of the precursor at the stage of calcination of the precipitate. This feature makes it possible to significantly reduce the evaporation of gallium oxide due to gallium bounding in a more complex compound. Figure 5 shows SEM images of the precipitate dried at 100 °C and then calcined at temperatures of 900, 1000, and 1100 °C for 2 h. SEM images show that at 900 °C agglomerates are formed from particles of the garnet habitus. Grain growth is observed with an increase in the annealing temperature of the precursors above 1000 °C. The choice heat treatment temperature of powder depends on the composition of the compound and should be experimentally selected. Definitely, it has to ensure the release of pores at the minimum acceptable sintering temperature of the compacted powder.

Worth noting, the formation of grains of garnet habitus in precursor obtained by spray pyrolysis is not observed even at a temperature of 1050 °C [110,111,122]. Spherical shape of the particles is retained by surface tension at high temperatures but does not prevent the evaporation of gallium. At the same time, the spray pyrolysis method is a well-established technology for the synthesis of nanopowders with a high specific surface area and low agglomeration. Precursors obtained by this method do not require additional milling and can be used for compaction without additional processing.

Several modifications of the co-precipitation method have been developed for disordered compounds with a garnet structure [107,123,124,125]. Among them, the method of reverse coprecipitation, which can significantly reduce the duration of the process of ceramic production is notable. The hydroxocarbonate precursor obtaining process includes four stages: (1) preparation of two solutions: a precipitant—ammonium bicarbonate and mixed metals solution; (2) precipitation of precursor by adding a mixture solution to the precipitant; (3), (4) washing and drying of the precipitate precursor. At the first stage solutions of metals nitrates are mixed in the required ratio and diluted to a concentration 0.5 M. Wherein the accuracy of concentration measurement should be ±0.1 g Me/kg of solution. The mixture solution was added in a thin stream with a flow rate of 40–60 mL/min to a precipitant with a concentration of about 1.5 M with continuous stirring, formed slurry was separated from mother liquor under vacuum. Then the precipitate was thoroughly washed with a water–isopropanol mixture. The product was dried to constant weight in a forced convection oven at 80–100 °C and then subjected to heat treatment at 800–1000 °C for 2 h to decompose the carbonates and form the garnet phase powder. Depending on the further compacting technology, the powder after low-temperature treatment is either used for milling and pressing, or it is subjected to a secondary heat treatment at a higher temperature to reduce its specific surface area and prepare it for compaction by colloidal methods.

### 3.2. 3D Printing Advances in Precursors Compacting

Consolidation of precursors under pressure remains the main method in the production of dense scintillation ceramics. At the same time, 3D printing of crystalline compounds with cubic space group with further annealing is becoming increasingly popular for the production of luminescent, laser, and scintillation materials. Among the promising methods of 3D printing, direct ink writing (DIW) might provide promising results [126,127,128,129,130,131]. DIW is an extrusion-based additive manufacturing method widely used to fabricate solid and/or mesh-like periodic structures. The DIW method has several advantages, namely its versatility, relative simplicity, and low cost of equipment. In the DIW method, “ink” is dispensed from small nozzles with variable-speed and deposited along digitally defined paths to produce planar or relatively simple 3D structures layer-by-layer. The rheological characteristics of such suspensions do not depend on the chemical nature of filler powder, its optical or other functional properties, but on its dispersion, the nature and concentration of a dispersant used, and the type of organic binder. The main limitation of the DIW method is the size of nozzles. Small nozzles provide relatively good resolution but result in relatively slow print speeds. The typical nozzle size used is 0.50–0.61 mm. The formulations described include sintering additives such as TEOS (tetraethyl orthosilicate) and MgO. To obtain dense ceramics and composites from nominally pure or doped YAG, mixtures of commercial simple oxides or ready-made commercial YAG nanopowders are used. Sintering is carried out, as a rule, at high temperatures of 1750–1850 °C under high vacuum conditions. Additionally, cold isostatic pressing (CIP) and hot-isostatic pressing (HIP) can be utilized.

In addition, we note another promising method—Stereolithography (or Vat photopolymerization) [132,133,134,135,136,137,138,139,140,141]. The method of stereolithography is based on the layer-by-layer controlled polymerization of photosensitive monomers (or suspensions with ceramic powders) under light irradiation. In 2002, the production of a regular columnar structure (a 9-rod array) from low-viscosity suspensions by stereolithography was demonstrated from composite acrylate polymers-Pb(Zr,Ti)O_3_ [138] with a typical size of a hundred microns. Recently, in 2020, the fabrication of a similar structure was demonstrated from another related dielectric ceramic high-Z material, BaTiO_3_ [139].

The first 3D printed complex inorganic polycrystalline YAG:Ce scintillator was successfully obtained by stereolithography in 2017 [132], and afterwards, the possibility of manufacturing transparent YAG:Yb ceramic was demonstrated [133]. Figure 6 shows an example of successful stereolithographic 3D printing of plate-shaped samples. The picture on the left panel is as-printed composites consisting of a mixture of Al_2_O_3_, Y_2_O_3_, Yb_2_O_3_, TEOS ceramic powders and an acrylate polymer(s). The picture on the right shows a translucent YAG:Yb ceramic plate after solid-state reaction and sintering in a high vacuum furnace at 1750 °C for 16 h. High temperatures up to 1800 °C, vacuum conditions, and sintering aids (TEOS, MgO, etc.) allow to produce transparent ceramics using the stereolithography method as well.

Stereolithography produces surfaces with high accuracy, good quality and low roughness. Its disadvantages include obvious restrictions on the available wall thickness and the size of holes in product prototypes (both top and bottom). A bottleneck in stereolithography could be the light source in the 3D printer, whose irradiation may be absorbed by the doping ions in the ceramic powders. Therefore, direct stereolithographic 3D printing with powders of the garnets activated by RE ions or transition metals requires a careful selection of the light emitter.

The possibility of printing several materials in one product looks promising, as recently demonstrated [139,140]. Figure 7 shows an example of successful stereolithographic 3D printing from several ceramic materials at the same time in one sample. In the upper row, as-printed composites are present; in the lower row, ceramic samples after sintering are shown.

Figure 8 shows a mesh scintillator in the form of a cylinder, which can be used for measurements in a liquid or gas flow [142]. It was obtained by the 3D stereolithography method from the GAGG:Ce powder with further annealing at a temperature of 1600 °C for 2 h.

Such an approach will make it possible in the near future to manufacture high-precision structured scintillator detecting elements consisting entirely of ceramics. For example, translucent GYAGG:Ce or other suitable garnet-type material will be used to print discrete pixels interacting with radiation, whereas undoped white GYAGG fill interpixel space, as a reflector. It should be noted that two-photon 3D printing has spatial resolution at the level of a few microns [136], which will certainly require the use of exclusively nanosized initial powders for ceramic production.

### 3.3. Sintering Process and Polycrystalline Structure Formation

The final step of producing a ceramic material is the sintering process, which is the formation of a dense polycrystalline mass under high temperature. When carrying out the sintering process under atmospheric conditions (in the air) it is difficult to achieve a complete exit of pores from the volume of ceramics. The residual porosity remains at a level of 0.5–1.0%. Transparent ceramics are obtained by sintering under vacuum conditions [105,111,143,144,145]. An alternative approach, which is suitable for transparent ceramics production, is sintering in an oxygen atmosphere, which prevents the evaporation of gallium [107,146,147,148] or hot isostatic pressing [122]. Figure 9 compares the microstructure of GYAGG:Ce ceramics obtained in different sintering atmospheres.

To obtain the ceramics with a minimum number of residual pores, sintering additives are used, which, when combined with a precursor, can affect the sintering process [149] and references there.

As an example, Figure 10 shows samples of GYAGG:Ce (a, c) and GYAGG:Tb (b, d) ceramics obtained by sintering in an oxygen atmosphere. The duration of sintering in this case was less than 4 h at a temperature of 1650 °C.

## 4. Disordered Doped Scintillation Materials

### 4.1. Heavy and Light Eu-Doped Scintillation Materials

Historically, the first scintillator doped with RE, the divalent europium ions, that found application was strontium iodide SrI_2_:Eu [150]. Material has been revisited in the last decade, and new halide compositions have been invited [151,152]. Several new alkali halide crystals have demonstrated impressive energy resolution parameters. Such materials, like other alkali halide compounds, were obtained in a single crystalline form by various pulling methods. The creation of a series of solid solutions [153] was a further development of this work. It was discovered that simultaneous incorporation of the heavy halide anions (I^−^, Br^−^) into the composition provides the best scintillation yield. We also note the outstanding characteristics of a mixture of bromide and barium iodide [154,155,156,157], as well as the preparation of BaBrCl compound by the Czochralski method [157], which indicates the prospect of producing a large ingot. An important improvement from mixing Cl and Br in an alkali halide composition is the reduction in the hygroscopicity of the compound. However, when mixing light and heavy halides in the lattice, no significant change in the scintillation yield was found [158]. The positive role of codoping with Au ions has been established [159,160] as well.

The influence of anion substitution in the lattice of an alkali-halide compound might be considered by analyzing the density of states in single-anionic compounds (Figure 11). Since the valence bands in BaI_2_, BaBr_2,_ and BaCl_2_ crystals are formed dominantly by the p-states of I^−^, Br^−^ and Cl-ions, the substitution disorder in mixed BaI_2(1-x)_Br_2x_ and BaBr_2(1-x)_Cl_2x_ crystals affects the states of the valence band, i.e., it leads to the scattering and localization of nonequilibrium holes. However, the effective mass of holes in the valence band of the investigated ionic crystals is quite large. Therefore, nonequilibrium holes have a small diffusion length and scattering due to the compositional disorder further limiting it. At the same time, nonequilibrium electrons are still more mobile than holes, which leads to the strong spatial dispersion of carriers. This situation is typical for BaBr_2(1-x)_Cl_2x_, in which the binary components forming the solid solution are direct bandgap crystals. The light yield of such an anionic mixture is not expected to be changed.

On the contrary, the mixed BaI_2(1-x)_Br_2x_ crystal has a peculiarity in the valence band when considering this system in the virtual crystal (VC) approximation, described in [75]. BaI_2_ is an indirect bandgap crystal. That leads to the degeneration of the highest energy state in the valence band in the mixed system BaI_2(1-x)_Br_2x_. Although fluctuations of the effective potential due to the compositional disorder destroy this degeneracy, there are two states of similar energy relating to different regions in the cell. However, the probability of hole tunneling between these states may be significant due to the strong uncertainty of the quasi-momentum in a mixed system. This effect might lead to an increase in the diffusion length of holes. Therefore, more efficient formation of excitons and, consequently, increased light yield are expected.

In addition to heavy Eu^2+^-doped scintillators, lithium-containing compounds containing atoms of alkaline earth elements (Ca, Sr), which are isovalently replaced by divalent europium, are of particular interest. Such compounds are light and have a practical demand in the field of thermal neutron detection when used in the form of scintillation coatings for neutron radiography setups. The “gold standards” scintillators of neutron sensitive screens are commercially available terbium-doped gadolinium oxysulfide Gd_2_O_2_S:Tb (GOS, Gadox) and ^6^LiF/ZnS:Ag mixtures. The advantages of these compounds include their high scintillation light yield and, in the case of GOS, also the possibility to get a high spatial resolution of screens, which is associated both with the short free path of conversion electrons in the material (~6 μm) and a high neutron cross-section making efficient thin scintillator layers [164,165]. However, the bottleneck of these scintillators is the long scintillation decay kinetics, which limits their use under high flux neutron beams. Furthermore, both have a large Z_eff_, which causes sensitivity to background gamma-quanta. This is more suitable for GOS, but for ^6^LiF/ZnS:Ag, the opacity of the ZnS to their scintillation light deteriorates spatial resolution of the composites as well as efficiency of light collection from a layer.

An innovative option to combine high light output, fast scintillation kinetics, and neutron absorption efficiency is a composite coating based on (Gd,Y)_3_(Al,Ga)_5_O_12_:Ce and ^6^LiF, which demonstrates a high response to neutrons at the level of commercial ^6^LiF/ZnS:Ag screens [166]. The light oxide lithium-containing compounds having a few cations seem to be prospective alternatives for GOS and ^6^LiF/ZnS:Ag.

The lithium aluminate LiAlO_2_ is well known among a variety of light lithium-containing binary compounds. However, the light yield of scintillations is low when doped with rare-earth and transition metal ions and does not exceed 9000 ph/neutron [167,168]. Light ternary compounds containing lithium and alkaline earth metals have also been studied quite well as phosphors of different luminescence colors. For scintillators, over the past year attempts have been made to grow crystals of relatively heavy lithium-containing eutectics: the lithium-strontium and lithium-barium chlorides [169,170]. An LiCaAlF_6_ scintillator doped with cerium [171] and divalent europium [172] is also known from non-oxide systems, but it has low scintillation yield: 9000–12,000 ph/neutron when doped with Eu^2+^. Among the three-component oxide systems containing Li_2_O and AE (where AE = Ca, Sr, Ba), one can mark those which contain boron oxide as the third component: LiCaBO_3_ [173] and a silicon oxide. Li_2_CaSiO_4_ (LCS) is one of the compounds with the least amount of silicon tetrahedra. The scintillation properties of cerium doped LCS were first described in [174], where a light yield of 7780 ph/MeV was reported upon excitation with gamma-quanta. In [175], the cathodoluminescence of Li_2_CaSiO_4_ doped with Eu^2+^ was studied. However, as applied to the detection of neutrons, this compound was first proposed by the authors of [176,177]. A thin (180 µm) two-component composite LCS layer demonstrated the same response to neutrons as a commercial 470 µm ^6^LiF/ZnS:Ag screen and a 14% lower response to gamma-quanta. The achieved light yield under 5.5 MeV alpha-particles was measured to be 21,600 photons/MeV, which corresponds to the yield under thermal neutrons of ~100,000 ph/neutron, which is associated with a fast scintillation kinetics equal to 500 ns. The measured properties are similar to those of known lithium-containing scintillators CLYC and CLLB [178] and, at the same time, they have the potential to further increase the scintillation light yield. One of the important factors affecting the luminescence intensity of Li_2_CaSiO_4_:Eu^2+^ is the localization of Eu^3+^ in the compound, which is a quenching agent of the Eu^2+^ luminescence. Oxidation of Eu^2+^ to Eu^3+^ occurs during synthesis due to the evaporation of lithium and charge compensation of the defects by oxygen vacancies. To stabilize europium in the divalent state, the authors of [179,180] proposed introducing the RE metal ions into the matrix. The occupation by trivalent RE ions of the positions of the divalent Ca^2+^ may compensate for the loss of Li^+^ by the following Equation (3):(3)$$V_{Li}^{\prime }+\frac{1}{2}V_{Ö}+Eu_{2}O_{3}\to V_{Li}^{\prime }+\frac{1}{2}V_{Ö}+Eu_{Ca}^{\times }\to V_{Li}^{\prime }+Eu_{Ca}^{.}+RE_{2}O_{3}\to V_{Li}^{\prime }+RE_{Ca}^{.}+Eu_{Ca}^{\times }\;\;$$

However, the incorporation of additional RE ions into Ca sites in the matrix increases the effective charge of the compound and thereby increases the sensitivity of the material to background gamma radiation. Another method of stabilizing Eu^2+:^ in the compound was found to be the substitution of Si ions by Al ions in tetrahedra. Both ions are light. The nonisovalent substitution of Si^4+^ by Al^3+^ ions act as a nonisovalent substitution in Ca sites in the compound. As a result, europium is dominantly stabilized in the divalent state according to the following Equation (4):(4)$$Al_{Si}^{\prime }=Al_{Si}^{\times }+e^{-}$$
(5)$$e^{-}+Eu_{Ca}^{.}\to Eu_{Ca}^{\times }$$

This technological approach to dilute Si cationic sublattice made it possible to increase the intensity of photoluminescence by a factor of two, which correlates with the proportions of Eu^2+^ and Eu^3+^ in the compound (Figure 12).

The relative amount of Eu^2+^ and Eu^3+^ in the compounds was estimated from the photoluminescence spectra at 395 nm excitation, which ensures the excitation of both Eu^3+^ and Eu^2+^ ions. It was estimated as the ratio of the integrated photoluminescence intensity of the characteristic Eu^2+^ peak at 480 nm and integrated photoluminescence intensity of the peaks at 593, 617, and 700 nm corresponding to Eu^3+^ ions to total integrated intensity in 430–730 nm spectral range (Table 1).

In the context of compositional disorder, it is also worth mentioning the partial substitution of O^2−^ ions by N^3−^ ions, as is realized in oxynitride phosphors [181]. The most studies aimed at improving the luminescence properties. However, the door is open for researchers to use the same techniques as a tool to improve the scintillation properties of desired groups of compounds.

### 4.2. Ce-Doped Materials

Introducing disorder into both the cationic and anionic subsystems in Ce-activated crystals also gave a new impetus to research on improving the scintillation properties of alkali halide and oxide scintillators. Some results, after technological experiments, could be explained in the way described above for Eu^2+^ doped halides. A solid solution of LaI_3_ and LaBr_3_ having a mixture of anions was found to have a smaller LY in a comparison with pure LaBr_3_. However, it demonstrated faster scintillation kinetics [182]. The significant shortening of the scintillation kinetics can be explained by the possible contribution of tunneling to hole mobility, as in disordered BaI_2(1-x)_Br_2x_. The decrease in the scintillation yield can be explained by scattering in a solid solution due to the difficulty of stabilizing one phase when mixing hexagonal and orthorhombic phases [183]. The creation of disorder due to the substitution of cations in the compounds LuI_3_–YI_3_ or LuI_3_–GdI_3_ also does not lead to an increase in the yield of scintillations [184]. This is also due to technological difficulties in obtaining a single-phase solid solution of these compounds; YI_3_ has a trigonal symmetry group, while GdI_3_ has a hexagonal one. All three iodides are indirect-gap compounds, which makes it difficult to describe the dynamics of nonequilibrium carriers in such mixed systems yet.

Studies of the scintillation properties of oxyhalides have been carried out by the authors [185]. The creation of disorder due to the inclusion of oxygen and halide ions as ligands in the composition led to a significant deterioration in the parameters. The samples were obtained in the form of powders by solid-phase synthesis. Apparently, the deterioration of the parameters is associated with the difficulty of obtaining a homogeneous single crystalline phase.

Doping the well-known BaF_2_ scintillation material with Ce ions results in a slowing of the kinetics but a significant increase in scintillation yield [186,187,188]. However, in the LaF_3_-CeF_3_ system, a cation mixture does not significantly improve the parameters [189]. Cs_2_HfCl_6_:Ce [190] is another recently discovered material worth noting. The material is not hygroscopic and has a cubic spatial symmetry. The introduction of disorder into the anionic subsystem does not increase light yield but does reduce the slow component of scintillation [191].

Summarizing the available experimental data, it can be concluded that the introduction of disorder by mixing direct-gap and indirect-gap alkali halide compounds can lead to improvements in some scintillation parameters, as is observed in the listed examples of Eu^2+^ and Ce^3+^ doped materials.

In contrast to alkali-halide compounds, the binary and ternary oxide compounds, which have the structure of garnet, perovskite, pyrosilicate or oxyorthosilicate, provide the formation of disordered compounds when mixed with compounds with the same space symmetry group. All of these compounds are also direct-gap. Therefore, the change in the scintillation parameters in such compounds exclusively occurs due to a change in the dynamics of nonequilibrium carriers in the conduction band and excitons as well. As noted above (Section 2.1), creating disorder in the Gd/Al subsystem creates the potential to increase the scintillation yield. The scintillation yield in GAGG:Ce ternary garnet can reach 51,000 ph/MeV [192,193,194], which is nearly 70% higher than in binary YAG:Ce or LuAG:Ce compositions [195]. On the contrary, when the modulation of the crystal potential is not large, as in the case of a combination of Y and Lu in the composition [196], Y and Gd in the composition (Y,Gd)_3_Al_5_O_12_ [197] and Lu and Gd [198], the increase in the scintillation yield is no more than twenty percent.

Here we underline the role of trivalent gadolinium ions in the crystal lattice [199]. Contrary to the crystals on a base of Y, La, Lu, which have no peculiarities of the electron density states into the forbidden zone, the Gd^3+^ based crystal host has numerous *f*-levels (P, I, D) creating sub-zones in the bandgap. Moreover, following to extended Dieke diagram [200], one can suppose that numerous Gd^3+^ levels in the energy ranger above 40,000 cm^−1^ will provide an effective capture of nonequilibrium carriers by Gd^3+^ ions and further intracenter relaxation to the lowest *6P* state. The intracenter relaxation process is quite short and carries on within a few picoseconds [6]. Consequently, the significant part of nonequilibrium carriers create Frenkel type excitons, which have a capability for migration along with P-, S-states of Gd sublattice. Moreover, holes can be quickly delocalized to the top of a valence band from 8S state [94]. Spatially, the hole can be localized in the same polyhedron with the mother Gd^3+^ ion, but predominantly located at the coordinating oxygen, and the Frenkel exciton might be considered as localized on the Gd-based oxyanionic complex. Migration of excitons between oxyanionic complexes provides delivery of excitation to cerium ions populating the Ce3+ radiating level 5d_1_.

The transition from ternary to quaternary garnets should take into account the breaking of the gadolinium sublattice and the features introduced in this way into the process of relaxation of nonequilibrium carriers. The authors of [192] found that in single crystals, such a transition, as a rule, leads to a lower scintillation yield. At the same time, in ceramics, where it becomes possible to use higher concentrations of the Ce^3+^ activator, it is possible to maintain or even exceed the scintillation yield in comparison with GAGG ceramics [201], while the scintillation kinetics can be significantly accelerated. An increase in the concentration of the activator, although it leads to the segregation of the activator near the grain boundaries [202], does not lead to a significant change in the scintillation parameters. There is a small fraction of such ions compared to those localized in the bulk of the grains. In a composition where some fraction of the Gd^3+^ ions are substituted by yttrium ions, it is possible to combine a high yield of 51,000 photons/MeV and fast scintillation kinetics (decay constant 50 ns) [203]. Of particular interest is the combination of Lu^3+^ and Gd^3+^ ions in a quaternary garnet to increase the stopping power of the material. At the same time, it was found that the partial substitution of gadolinium by lutetium ions leads to a significant decrease in the scintillation yield in both single crystal and polycrystalline samples compared to ternary GAGG [204,205,206]. Moreover, in such quaternary systems, a slow component appears in the scintillation. Breaking the integrity of the gadolinium sublattice by substitution with heavy lutetium ions increases the role of self-trapped states in the excitation of Ce^3+^ ions, which ensures both an increase in the fraction of short ~20 ns and very long ~600 ns components in the scintillation kinetics [207]. It is possible to preserve the fast kinetics, significantly reduce the slow kinetics, and increase the scintillation yield by changing from a quadruple to a quintuple composition by diluting the lutetium sublattice with yttrium ions. The ceramics sample with the composition Gd_2_Y_0.5_Lu_0.5_Al_2_Ga_3_O_12_ doped with Ce^3+^ and codoped with a small Mg^2+^ concentration (20 ppm) demonstrates fast scintillation kinetics having components of 14 ns (84%); 78 ns (16%) and the light yield at the level of 41,000 ph/MeV [208]. Scintillation materials based on disordered compounds with a garnet structure have come close in their parameters to advanced alkali-halide materials, surpassing them in stopping power as well as in manufacturability.

In oxyorthosilicates, when turning from Lu_2_SiO_5_ to (Lu,Y)_2_SiO_5_, it is hard to improve the scintillation yield due to the small depth of modulation at the bottom of the conduction band (Section 2.1). However, the introduction of yttrium into the composition allows solving the problem of phase homogeneity as well as minimizing the role of deep traps in capturing nonequilibrium electrons, that is, reducing the level of phosphorescence.

In pyrosilicates, mixing lanthanum and gadolinium ions (La,Gd)_2_Si_2_O_7_:Ce [209,210] in the composition makes it possible to maintain high scintillation yields at high temperatures and increase the yield by 20–25% compared to Gd_2_Si_2_O_7_:Ce.

In halide perovskites [211], it is not yet possible to obtain better results than for end compounds, while in (Lu, Gd)AlO_3_:Ce [212] the light yield approaches 21,000 ph/MeV, which is close to that of but much less dense YAlO_3_:Ce and 30% higher than the best (Lu,Y)AlO_3_:Ce. The compound may reach a Z_eff_~65, which is comparable to commercial LSO:Ce. Nonetheless, its scintillation kinetics are found to be three times slower when compared to YAlO_3_: Ce.

### 4.3. Tb Doped Materials

Terbium Tb^3+^ ion is another activator for creating bright scintillators. It can be hardly used for pulse height spectra acquisition due to its long scintillation kinetics (~2 ms), but can be utilized in a current mote with sensitive photo-diodes. Widely used in X-ray scanners is Gd_2_O_2_S translucent ceramics, activated with Tb^3+^ ions [213]. The host compounds for activation with terbium ions, which can be created in a polycrystalline form, are of particular interest [214]. A high concentration of the Tb activator is achievable compared to Ce activator in garnet type crystals [215]. This can significantly expand the field of utilization of such scintillation materials, for example, in alpha-, beta-voltaics [216], where the efficiency of energy conversion of ionizing radiation can reach or even exceed the efficiency of thermoelectric generators or cathodoluminescent light sources [217]. Moreover, due to high neutron cross-section, cerium doped Gd-based ternary garnets were shown to be promising scintillation materials to detect neutrons in a wide energy range [218,219]. Terbium provides slower but brighter scintillation than cerium, and this makes terbium-activated gadolinium-containing materials promising for application in scintillation threshold neutron counters or neutron sensitive screens, where sensitivity is the most important property. Figure 13a shows SEM image Tb-doped (Gd_,_Y)_3_Al_2_Ga_3_O_12_ ceramics; its microstructure is identical to those doped with Ce. Figure 13b demonstrates shining of Tb-doped ceramics under electron beam.

Worth noting, Tb-doped sample shows a small temperature coefficient for the light yield LY(T) in the temperature range 300–540 K. Quaternary Gd-Y compounds with a garnet structure doped with Tb were found to be a good platform to create high-bright scintillators to be exploited in indirect converters with both β- and α- isotopes. Distinctive features of the developed scintillators are their tolerance to irradiation and temperature stability, which are of especial importance for application in converters. The materials tested were fabricated as ceramics to utilize a flexibility with concentration of doping ions and to achieve high doping levels, which are hardly attained in the single crystalline matrix.

## 5. Conclusions

An extensive overview of the research and creation of scintillation crystal materials with compositionally disordered structures is provided. Both single- and poly-crystalline RE doped materials show an improvement in the scintillation properties. Amorphization of a chemical does not necessarily result from compositional disorder in a crystalline structure. A disordered distribution of ions occurs in the matrix when a polycationic ensemble with identical crystallographic sites forms in the cationic sublattice. The ordered distribution of anions, however, causes the space symmetry group to be conserved. As a result, crystallinity—or long-range order—in the ion stabilization locations is maintained. This approach is typical for oxide systems. For alkali-halides, where there is more latitude in the admixing, anions may be mixed as well. In this case, the long-range order is preserved due to the ordering of the cationic subsystem. The compositional disorder causes a modulation of the effective crystalline potential. Apparently, most of the positive effects on scintillation properties are due to the effects of the modulation of the effective potential in a disordered crystal on the mobility of nonequilibrium carriers. The modulation of the effective potential leads to a decrease in the diffusion length of free carriers, increasing the concentration of geminate electron-hole pairs and, consequently, increasing the transfer efficiency to the luminescent centers. A production cycle for multiunit oxide compositions with a garnet structure is considered in detail. Inverse coprecipitation of semiproducts, combined with a short annealing of the green bodies in the oxygen atmosphere at a temperature significantly lower than the melting point, could result in transparent materials doped with different RE ions. Future optimization of the production process chain, which includes engineering at the nano-scale level, might be based on significant progress in the last decade on the topic. In our opinion, more of the materials discussed will emerge as candidates for commercialization. Crystalline scintillators with disordered composition may become more prevalent in future ionizing radiation detectors due to the demand for new applications requiring purposeful tuning of the scintillation parameters.

## Figures and Tables

**Figure 1 nanomaterials-12-04295-f001:**
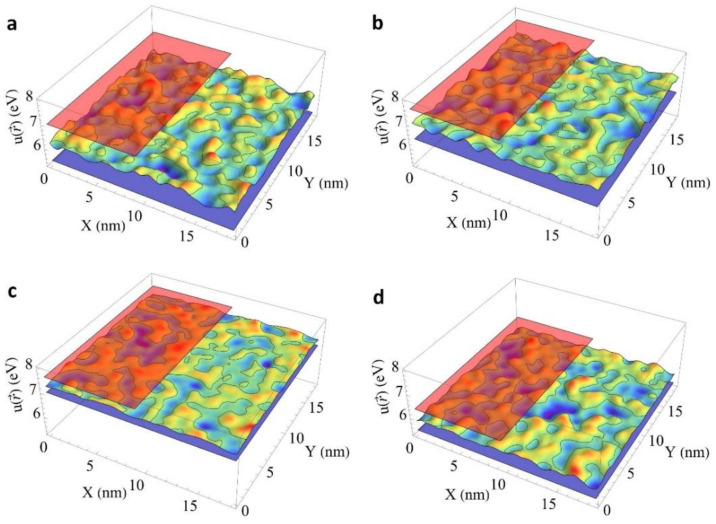
2D cross-sections of the spatial distribution of the pseudopotential $u\left(\vec{r}\right)$: (**a**) Y_3_Al_5_O_12_ (red plane, $E_{g}=7.01\;\mathrm{eV}$), Y_3_Ga_5_O_12_ (blue plane, $E_{g}=5.46\;\mathrm{eV}$) and Y_3_(Al,Ga)_5_O_12_; (**b**) Lu_3_Al_5_O_12_ (red plane, $E_{g}=7.61\;\mathrm{eV}$), Lu_3_Ga_5_O_12_ (blue plane, $E_{g}=6.38\;\mathrm{eV}$) and Lu_3_(Al,Ga)_5_O_12_; (**c**) Lu_3_Al_5_O_12_ (red plane), Y_3_Al_5_O_12_ (blue plane) and (Lu,Y)_3_Al_5_O_12_ and (**d**) Lu_3_Ga_5_O_12_ (red plane), Y_3_Ga_5_O_12_ (blue plane) and (Lu,Y)_3_Ga_5_O_12_.

**Figure 2 nanomaterials-12-04295-f002:**
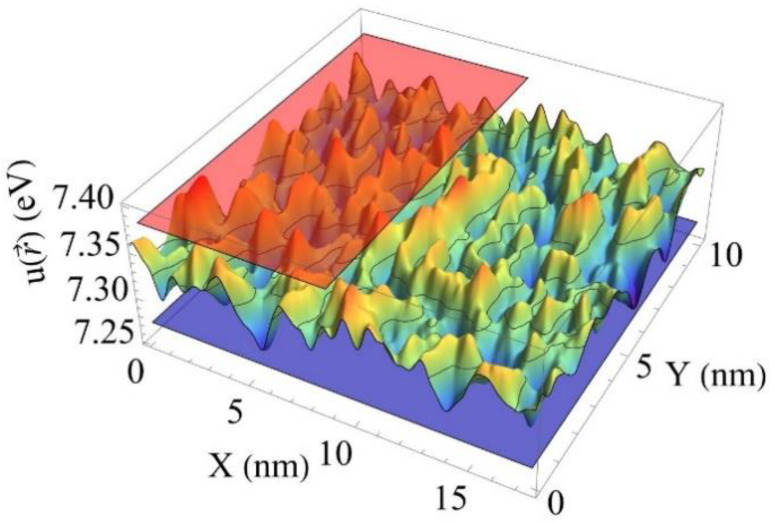
2D cross-sections of the spatial distribution of the pseudopotential $u\left(\vec{r}\right)$ in Lu_2_SiO_5_ (blue plane, $E_{g}=7.27\;\mathrm{eV}$), Y_2_SiO_5_ (red plane, $E_{g}=7.38\;\mathrm{eV}$) and (L_0.5_-Y_0.5_)_2_SiO_5_.

**Figure 3 nanomaterials-12-04295-f003:**
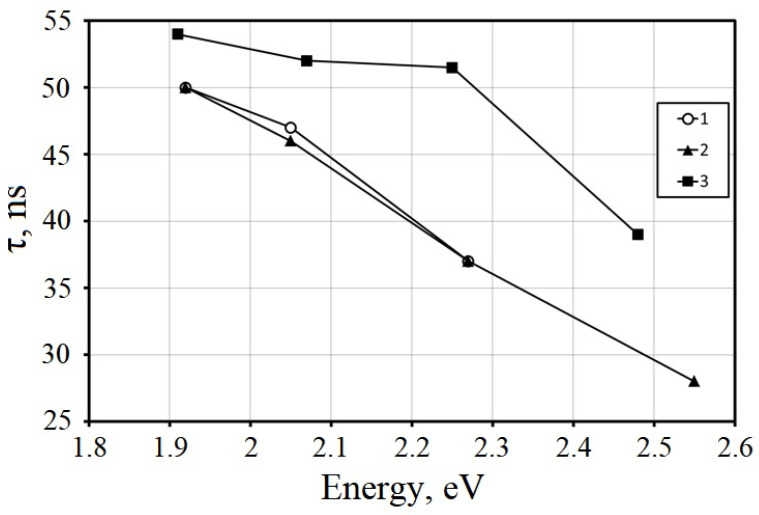
A change in the decay constant of the initial stage of the kinetics (a decrease by a factor *e*) of the photoluminescence of the Gd_1.495_Lu_1.495_Ce_0.015_ Ga_3_Al_2_O_12_ upon excitation 2.78 eV (1) and 3.67 eV (2) in comparison to Gd_2.985_Ce_0.015_Ga_3_Al_2_O_12_ upon excitation 2.78 eV (3).

**Figure 4 nanomaterials-12-04295-f004:**
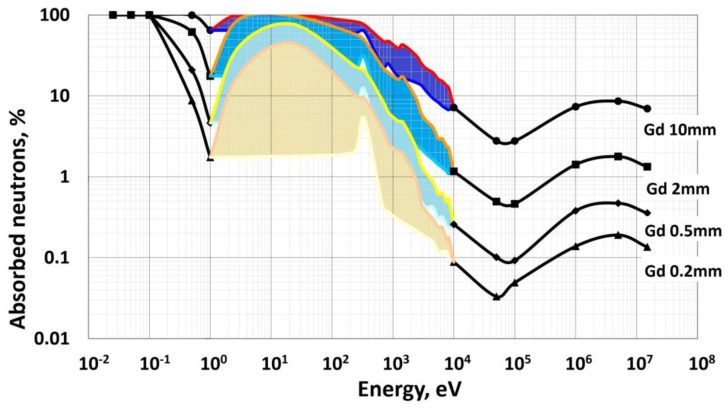
Percentage of absorbed neutrons in gadolinium metal of natural isotopic composition depending on the particle energy. The region of resonances is highlighted in color.

**Figure 5 nanomaterials-12-04295-f005:**
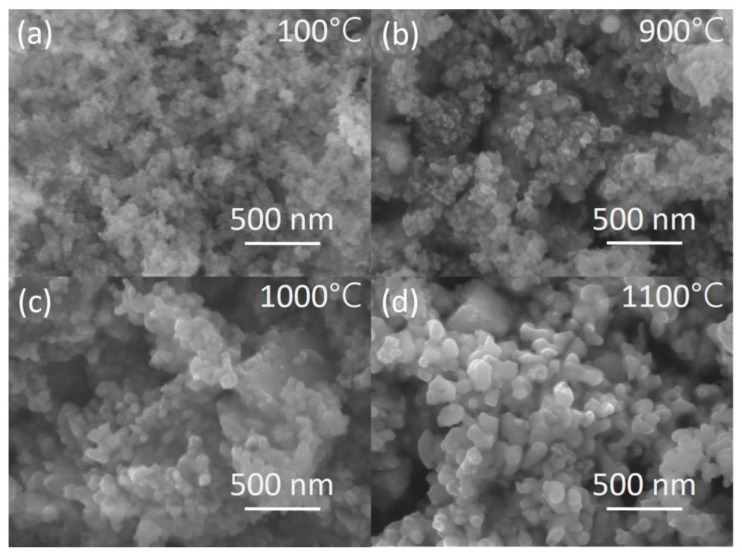
SEM images of GYAGG:Ce nanopowders calcined at 100 °C (**a**), 900 °C (**b**), 1000 °C (**c**), 1100 °C (**d**).

**Figure 6 nanomaterials-12-04295-f006:**
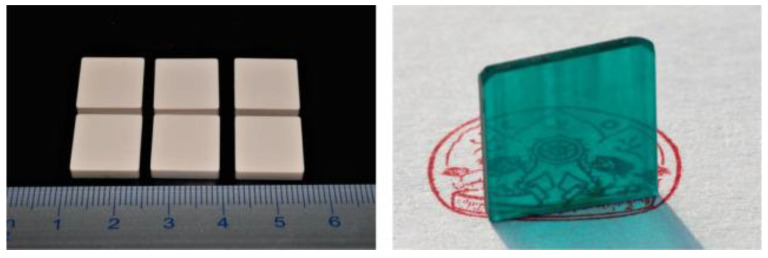
Photograph of as-printed samples (**left**) and of a polished sintered Yb:YAG ceramic sample after annealing (**right**). After [133].

**Figure 7 nanomaterials-12-04295-f007:**
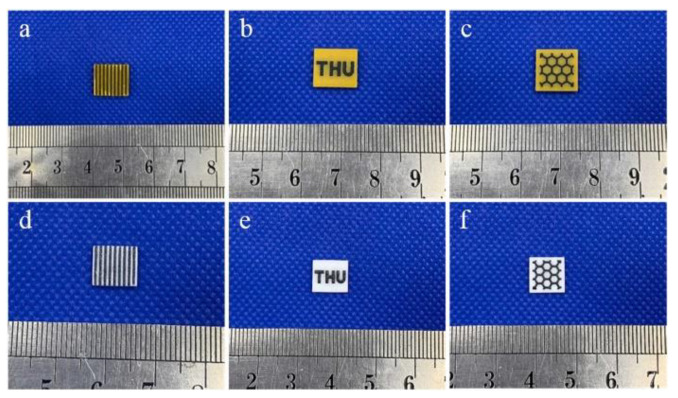
Photograph of as-printed green bodies (**a**–**c**) and sintered parts (**d**–**f**). After [140].

**Figure 8 nanomaterials-12-04295-f008:**
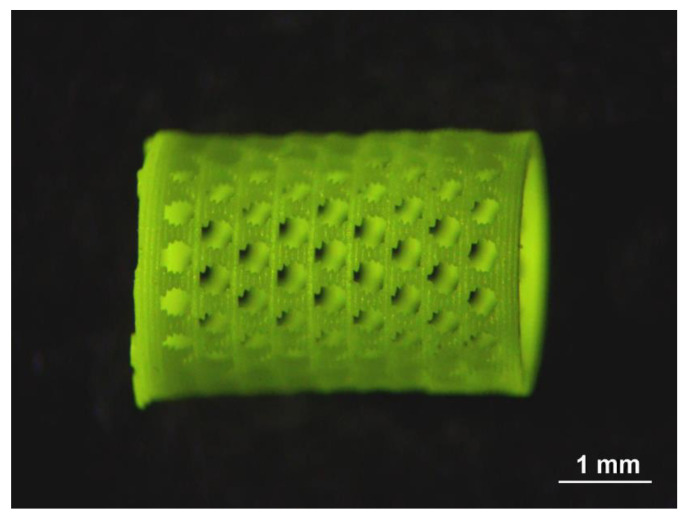
The mesh scintillator in the form of a cylinder obtained by the 3D stereolithography method from the GAGG:Ce powder with further annealing at a temperature of 1600 °C for 2 h. After [142].

**Figure 9 nanomaterials-12-04295-f009:**
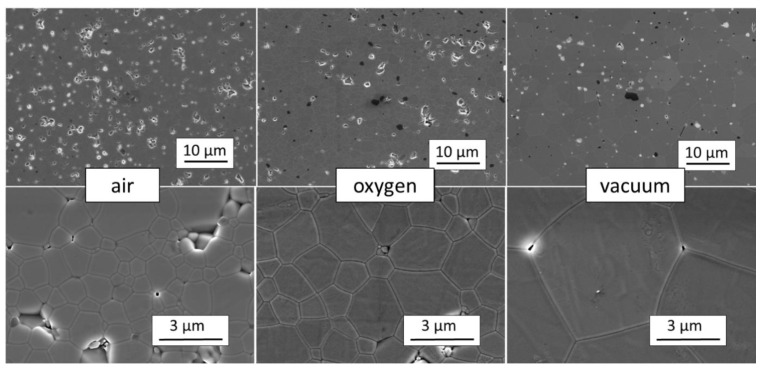
SEM images of ceramic samples obtained in different sintering atmospheres and vacuum.

**Figure 10 nanomaterials-12-04295-f010:**
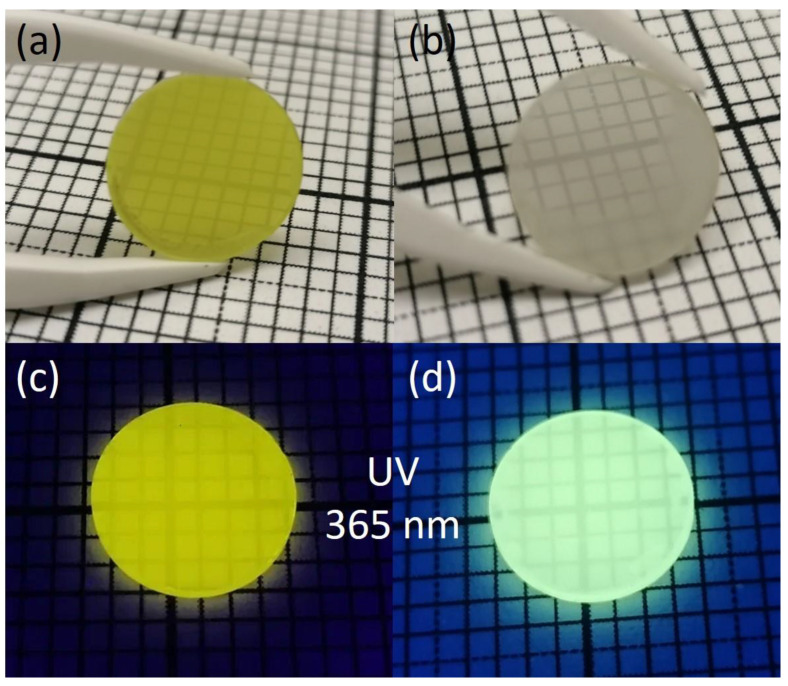
Images of ceramic samples GYAGG:Ce (**a**,**c**) and GYAGG:Tb (**b**,**d**) under natural light and under UV-light excitation (λ = 365 nm). The ceramic samples were double side polished.

**Figure 11 nanomaterials-12-04295-f011:**
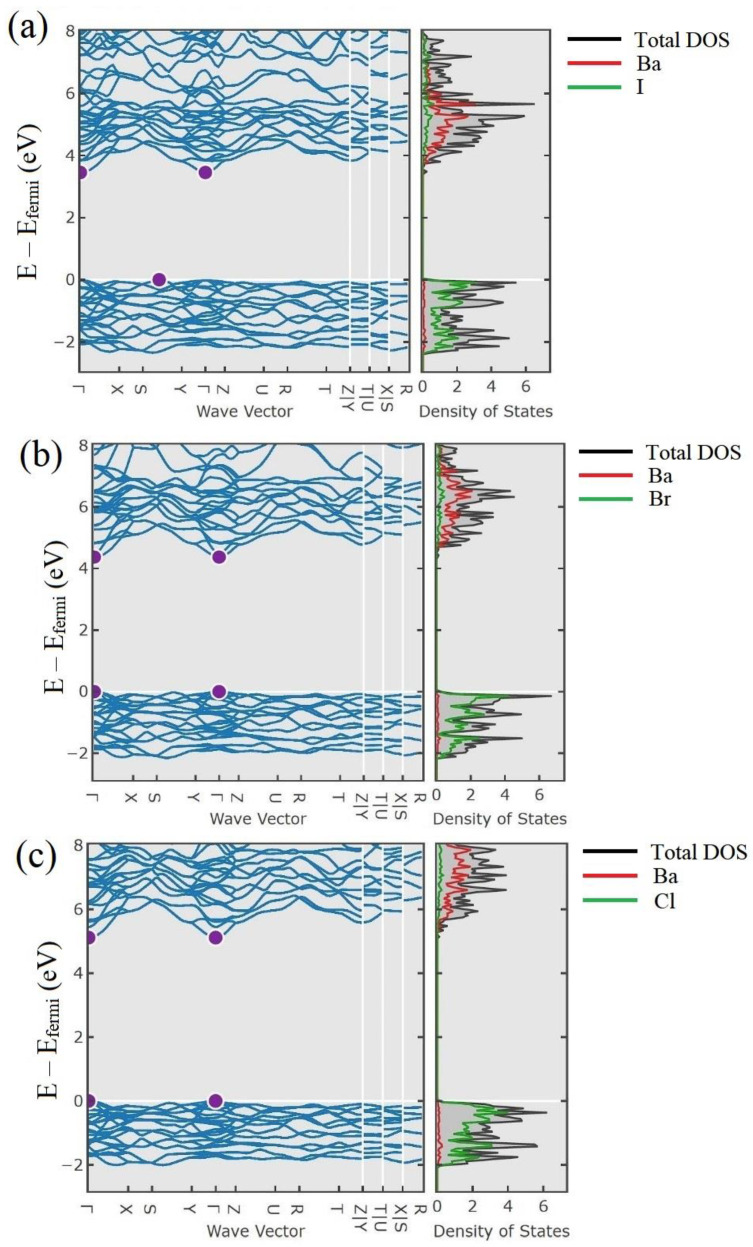
Band structure and the density of states: (**a**) BaI_2_ [161]; (**b**) BaBr_2_ [162]; (**c**) BaCl_2_ [163] crystals.

**Figure 12 nanomaterials-12-04295-f012:**
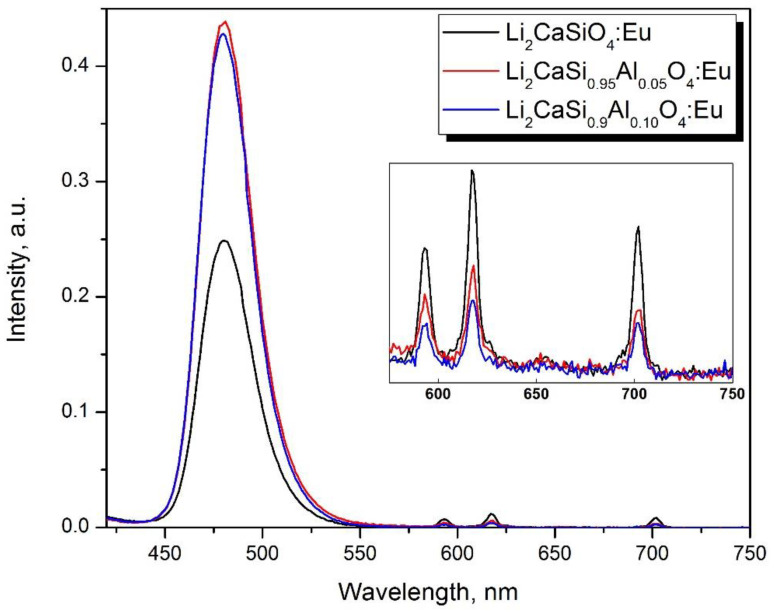
Photoluminescence spectra of Li_2_CaSiO_4_ samples and samples with compositional disordering in Si-sublattice obtained by partial Al substitution registered at excitation wavelength of 395 nm.

**Figure 13 nanomaterials-12-04295-f013:**
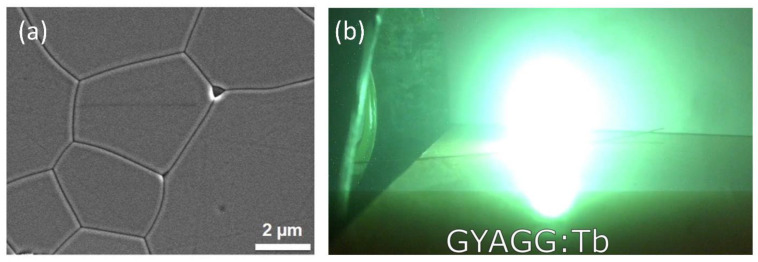
Scanning electron microscopy image of Gd_1.13_Y_1.72_Tb_0.15_Al_2_Ga_3_O_12_ sample (**a**), this sample under excitation with 150 keV electron beam, current density 1A/ cm^2^ (**b**). After [220].

**Table 1 nanomaterials-12-04295-t001:** The fraction of the luminescence of Eu^2+^ and Eu^3+^.

Composition.	Eu^2+^%	Eu^3+^%
Li_2_CaSiO_4_: Eu	98.7 ± 0.1	1.3 ± 0.1
Li_2_CaSi_0.95_Al_0.05_O_4_: Eu	99.5 ± 0.1	0.5 ± 0.1
Li_2_CaSi_0.90_Al_0.10_O_4_: Eu	99.6 ± 0.1	0.4 ± 0.1

## Data Availability

Not applicable.

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
