# Peer review of "Compositionally Disordered Crystalline Compounds for Next Generation of Radiation Detectors"

_nanomaterials, 2022, doi:10.3390/nano12234295_

Round 1
Reviewer 1 Report
Hi, Authors.
The review is interest and useful. I recommend to accept the manuscript after minor corrections.
1. The review of Ce-doped materials exclude the BaF2:Ce material as single crystal and optical ceramics. I think what the describing advantages and disadventages of this material will be improved the manuscript.
2. In line (215) used #x# without describing meaning. What is the means the #k - scale factor#?
3. The discussion concerning disordering exclude the phonon mean free path and thermodynamic characteristics which are usefull for estimation of promising scintillators as described in [L. Dumoulin et al. Journal of Mater Chem C 2022. 10, 5218 - 5229 doi. 10.1039/D1TC06166F].
4. The quality of Fig.3, 11 need to improved.
5. The manuscript contain typos. Line 747, the transition is not correct. 2F5/2-2F7/2. Line 793. #Tb doped materials# rather than #T. b doped materials#. Line 794. Second proposal start from #I#. I think what is typo and better start from #It#.
Cheers.
Author Response
The review is interest and useful. I recommend to accept the manuscript after minor corrections.
- The review of Ce-doped materials exclude the BaF2:Ce material as single crystal and optical ceramics. I think what the describing advantages and disadventages of this material will be improved the manuscript.
Authors response.
We have added an appropriate description and five new references as well.
Doping the well-known BaF2 scintillation material with Ce ions results in a slowing of the kinetics but a significant increase in scintillation yield [186, 187]. However, in the LaF3-CeF3 system, a cation mixture does not significantly improve the parameters [188]. Cs2HfCl6:Ce [189] is another recently discovered material worth noting. The material is not hugroscopic and has a cubic spatial symmetry. The introduction of disorder into the anionic subsystem does not increase light yield but does reduce the slow component of scintillation [190].
Wojtowicz, A.; Szupryczynski, P.; Glodo, J.; Drozdowski, W.; Wisniewski, D. Radioluminescence and recombination process in BaF2:Ce. J. Phys. Condens. Matter 2000, 12, 4097, doi:10.1088/0953-8984/12/17/315.- Luo, J.; Sahi, S.; Groza, M.; Wang, Z.; Ma, L.; Chen, W.; Burger, A.; Kenarangui, R.; Sham, T.-K.; Selim F.A. Luminescence and scintillation properties of BaF2Ce transparent ceramics. Optical Materials2016, 58, 353-356, doi:10.1016/j.optmat.2026.05.059.
- Auffray, E.; Baccaro, S.; Beckers, T.; et al. Extensive studies on CeF3 crystals, a good candidate for electromagnetic calorimetry at future accelerators, NIM A 1996, 388, 367-390, doi:10.1016/S0168-9002(96)00806-6.
- Burger, A.; Rowe, E.; Groza, M. et al. Cesium hafnium chloride: A high light yield, non-hygroscopic cubic crystal scintillator for gamma spectroscopy, Appl. Phys. Lett.2015, 107, 143505, doi:10.1063/1.4932570.
- Hawrami, R; Ariesanti, E.; Buliga, V.; Matei, L.; Motakef, S.; Burger A. Advanced high-performance large diameter Cs2HfCl6 (CHC) and mixed halides scintillator, Journal of Crystal Growth, 2020, 533, 125473, doi: 10.1016/j.jcrysgro.2019.125473.
- In line (215) used #x# without describing meaning. What is the means the #k - scale factor#?
Corrected, meaning is included.
- The discussion concerning disordering exclude the phonon mean free path and thermodynamic characteristics which are usefull for estimation of promising scintillators as described in [L. Dumoulin et al.Journal of Mater Chem C 2022. 10, 5218 - 5229 doi. 1039/D1TC06166F].
Authors response.
The authors thank the Reviewer for the reference of just published article. We have considered an additional limitation of the mobility of free electrons caused by elastic scattering on pseudopotential fluctuations. Of course, the scattering caused by the emission and absorption of LO phonons mainly determines the mobility of electrons in an ionic crystal, but the phonon scattering rates in the application of a virtual crystal and in the approximation taking into account fluctuations of the crystal potential are close. Here we have limited ourselves to only considering such elastic scattering.
- The quality of Fig.3, 11 need to improved.
Authors response.
The figures have been improved in quality and are now in.jpg format.
- The manuscript contain typos. Line 747, the transition is not correct. 2F5/2-2F7/2. Line 793. #Tb doped materials# rather than #T. b doped materials#. Line 794. Second proposal start from #I#. I think what is typo and better start from #It#.
Authors response.
Corrected.

Reviewer 2 Report
This is a well-written review paper on compositional disordered crystalline compounds. There are just a few minor issues that may need clarification before publication.
1) Introduction, page 2: "Since the publication of the second edition of the book [7]...". Which book is referenced here? Please be more descriptive. Say something like "...the second edition of the book on Inorganic Scintillator for Detectors Systems by LeCoq et al.".
2) Conduction band bottom landscape modulation, page 5: "The difference in the formation of the landscape at the bottom of the conduction band..." It is unclear to what "landscape" refers to. Please use more scientifically descriptive words such as 'dispersion' or at a minimum explain what "landscape" is.
3) Modulation of local charge distribution, page 8: It is interesting the authors mention gadolinium as an ion in garnets for neutron detection. However, would substitution with boron not be of interested either? I do not see this mentioned in the paper. Could the authors please comment on substituting Al or Ga with B and its effect on properties?
Author Response
This is a well-written review paper on compositional disordered crystalline compounds. There are just a few minor issues that may need clarification before publication.
- Introduction, page 2: "Since the publication of the second edition of the book [7]...". Which book is referenced here? Please be more descriptive. Say something like "...the second edition of the book on Inorganic Scintillator forDetectors Systems by LeCoq et al.".
Authors response.
Corrected.
- Conduction band bottom landscape modulation, page 5: "The difference in the formation of the landscape at the bottom of the conduction band..." It is unclear to what "landscape" refers to. Please use more scientifically descriptive words such as 'dispersion' or at a minimum explain what "landscape" is.
Authors response.
We agree with the Reviewer. The "the landscape at ..." is replaced with "the spatial distribution of ...".
- Modulation of local charge distribution, page 8: It is interesting the authors mention gadolinium as an ion in garnets for neutron detection. However, would substitution with boron not be of interested either? I do not see this mentioned in the paper. Could the authors please comment on substituting Al or Ga with B and its effect on properties?
Authors response.
We agree with the reviewer that it could be interesting. However, referring to ionic radii data, one can see that the difference between B3+, Al3+, Ga3+ radii is too large. In a garnet-type compound, the Al3+ and Ga3+ ions occupy both tetrahedral and octahedral coordinations. The radii values of B3+, Al3+, Ga3+ ions are: 0.11, 0.39, 0.47 Å and 0.27, 0.54, 0.62 Å for tetrahedral and octahedral positions, respectively. While the difference in ionic radii should not exceed 10-15% for substitution in the crystal lattice to keep a single phase . Furthermore, only three boron-containing compounds with Ia3d symmetry, which might be pormissing, are known: Sr4Li(BO3)3, NaSr4(BO3)3, and Ba4Na(BO3)3. They are out the scope of the manuscript.

Round 2
Reviewer 1 Report
Hi,
Authors,
The Answer is not full and it has a new typo.
1. The choosing of references concerning BaF2:Ce is not enough because excluded papers Professors P.A. Rodnyi and P.P Fedorov, which investigated the BaF2:Ce optical ceramics before paper [187], for example, [Physics of the Solid State. volume 52, pages 1910–1914 (2010). doi.org/10.1134/S1063783410090209].
2. New text in paper contained a typo in word #hygroscopic# instead of #hugroscopic# in proposal #The material is not hugroscopic and has a cubic spatial symmetry.#
With best regards,
Reviewer.
Author Response
- The choosing of references concerning BaF2:Ce is not enough because excluded papers Professors P.A. Rodnyi and P.P Fedorov, which investigated the BaF2:Ce optical ceramics before paper [187], for example, [Physics of the Solid State. volume 52, pages 1910–1914 (2010). doi.org/10.1134/S1063783410090209].
Author response.
We agree with the Reviewer. This article was published first, it is included in the reference list.
New text in paper contained a typo in word #hygroscopic# instead of #hugroscopic# in proposal #The material is not hugroscopic and has a cubic spatial symmetry.#
Author response
Corrected.
